# Orthovoltage X-ray Minibeam Radiation Therapy for the Treatment of Ocular Tumours—An In Silico Evaluation

**DOI:** 10.3390/cancers15030679

**Published:** 2023-01-21

**Authors:** Tim Schneider, Denis Malaise, Frédéric Pouzoulet, Yolanda Prezado

**Affiliations:** 1Institut Curie, Université Paris-Saclay, CNRS UMR3347, Inserm U1021, Signalisation Radiobiologie et Cancer, 91400 Orsay, France; 2Department of Ophthalmology, Institut Curie, 75005 Paris, France; 3LITO, INSERM U1288, Institut Curie, PSL University, 91898 Orsay, France; 4Département de Recherche Translationnelle, CurieCoreTech-Experimental Radiotherapy (RadeXp), Institut Curie, PSL University, 91400 Orsay, France

**Keywords:** orthovoltage, X-rays, minibeams, MBRT, ocular tumours, Monte Carlo simulation, in silico, cost-effective

## Abstract

**Simple Summary:**

Radiotherapeutic treatments of ocular tumours are often challenging due to nearby organs at risk and the high doses required to treat radioresistant cancers such as uveal melanomas. Despite advanced techniques such as proton therapy and stereotactic radiosurgery, side effects in structures such as the lens, eyelids or anterior chamber remain an issue. Minibeam radiation therapy (MBRT) could represent a promising alternative in this context: MBRT is an innovative treatment approach based on spatial fractionation of the dose and sub-millimetre beam sizes that has been shown to improve normal tissue sparing while maintaining high tumour control. In this proof-of-concept study, we performed Monte Carlo simulations to evaluate MBRT with orthovoltage X-rays as a cost-effective treatment alternative for ocular tumours. The obtained doses were comparable to those reported in previous X-ray MBRT experiments and encourage the realisation of dedicated animal studies.

**Abstract:**

(1) Background: Radiotherapeutic treatments of ocular tumors are often challenging because of nearby radiosensitive structures and the high doses required to treat radioresistant cancers such as uveal melanomas. Although increased local control rates can be obtained with advanced techniques such as proton therapy and stereotactic radiosurgery, these modalities are not always accessible to patients (due to high costs or low availability) and side effects in structures such as the lens, eyelids or anterior chamber remain an issue. Minibeam radiation therapy (MBRT) could represent a promising alternative in this regard. MBRT is an innovative new treatment approach where the irradiation field is composed of multiple sub-millimetric beamlets, spaced apart by a few millimetres. This creates a so-called spatial fractionation of the dose which, in small animal experiments, has been shown to increase normal tissue sparing while simultaneously providing high tumour control rates. Moreover, MBRT with orthovoltage X-rays could be easily implemented in widely available and comparably inexpensive irradiation platforms. (2) Methods: Monte Carlo simulations were performed using the TOPAS toolkit to evaluate orthovoltage X-ray MBRT as a potential alternative for treating ocular tumours. Dose distributions were simulated in CT images of a human head, considering six different irradiation configurations. (3) Results: The mean, peak and valley doses were assessed in a generic target region and in different organs at risk. The obtained doses were comparable to those reported in previous X-ray MBRT animal studies where good normal tissue sparing and tumour control (rat glioma models) were found. (4) Conclusions: A proof-of-concept study for the application of orthovoltage X-ray MBRT to ocular tumours was performed. The simulation results encourage the realisation of dedicated animal studies considering minibeam irradiations of the eye to specifically assess ocular and orbital toxicities as well as tumour response. If proven successful, orthovoltage X-ray minibeams could become a cost-effective treatment alternative, in particular for developing countries.

## 1. Introduction

Radiation therapy (RT) represents one of the main modalities for treating ocular tumors today [1,2]. Challenging cases such as radioresistant uveal melanoma can be treated with brachytherapy or advanced external beam techniques like proton therapy and stereotactic RT [3] which allow it to achieve local 5-year tumour control rates of up to 96% [4,5]. Due to the targeted dose delivery and the steep dose gradients achievable with these high-precision modalities, vision-related structures as the fovea, the optic nerve or the optic chiasm can often be spared and the patient’s vision is usually preserved [5]. Nevertheless, side effects may be caused in the anteriorly positioned normal organs (such as lids, lens, iris, ciliary body and limbus) or, due to safety margins, in the retina around the tumor (possibly including the macula or optic nerve) which may entail a considerable impairment of visual acuity.

Minibeam radiation therapy (MBRT) is a promising new treatment approach that in preclinical studies has been shown to improve normal tissue sparing while maintaining high tumor control rates [6]. In the context of ocular tumours, it could help to further reduce side effects in the aforementioned organs at risk (OARs) and to better preserve the patient’s vision. In MBRT, the field is segmented into multiple narrow beamlets (called *minibeams*) that are spaced apart to create a highly modulated dose pattern characterized by alternating regions of high-dose (*peaks*) and low-dose (*valleys*) [6]. This so-called *spatial fractionation* of the dose represents a sharp contrast to conventional RT where a broad solid beam is used (usually widths >1 cm) to deliver a laterally homogeneous dose distribution. Delivering the dose in this way has been shown to drastically improve the tolerance of normal tissues with brain and skin tissue in rodents withstanding peak doses as high as 100–150 Gy [7,8,9]. This, in turn, may allow a safe escalation of the target dose and thus a more effective treatment of radioresistant tumours. Typical beamlet sizes in MBRT range from 0.1 to 1 mm full width at half maximum while the beamlet spacing is usually between 1 and 4 mm center-to-center (ctc) [6,10].

In contrast to the megavoltage X-rays used in conventional RT, MBRT is ideally performed with orthovoltage beams as this allows it to reduce lateral scattering and to maintain a more favourable *peak-to-valley dose ratio* (PVDR) in depth [11]. In this context, Prezado et al. [12] recently demonstrated the implementation of orthovoltage X-ray MBRT at a small animal irradiator, the Small Animal Radiation Research Platform (SARRP) by Xstrahl (https://xstrahl.com/sarrp/ accessed on 16 January 2023) [13]. Similarly, Bazyar et al. [9] reported an implementation using the X-RAD-320 by Accela (https://www.accela.eu/precision-x-ray/x-rad-320 accessed on 16 January 2023). While both irradiators are conceived for preclinical use, similar clinical orthovoltage machines exist, such as the Xstrahl 200 (https://xstrahl.com/xstrahl-200/ accessed on 16 January 2023). A convenient advantage of such irradiators is furthermore their relatively small footprint and low costs, in particular when compared to proton therapy facilities or stereotactic RT installations.

Considering all of the above, orthovoltage X-ray MBRT could represent a potential, cost-effective alternative for the treatment of ocular tumors that might allow a further reduction of radiation side effects while maintaining high treatment efficacy. The aim of this study was to perform a first evaluation of this idea by means of Monte Carlo (MC) simulations. To the authors’ knowledge, this represents the first study considering X-ray MBRT in the context of ocular tumors.

## 2. Materials and Methods

MC simulations were performed with the Geant4 based toolkit TOPAS (http://www.topasmc.org accessed on 16 January 2023) (version 3.7) [14,15] to assess the dose distributions in CT images of a human head under different irradiation conditions. Analogous to previous minibeam studies involving the SARRP irradiator [16,17], the simulated geometry comprised a virtual beam source, a collimator system and an irradiation target (human head CT images). The virtual beam source was based on the 220 kV spectrum of the SARRP, where, as in the study of Sotiropoulos et al. [17], photon energies were discretized in steps of 1 keV and contributions below 21 keV were omitted to improve the computational efficiency. A validation of the beam model has been carried out in previous studies [12,16]. Details concerning the minibeam collimator and the placement of the irradiation target are presented in the next section.

The physics list was built with the *Geant4_Modular* option using modules recommended for kilovoltage radiotherapy applications (*g4em-livermore*, *g4h-phy_QGSP_BIC_HP*, *g4decay*, *g4ion-binarycascade*, *g4h-elastic_HP* and *g4stopping*) [16,18,19]. A range cut of 1 μm was used in all cases and for all particle types.

### 2.1. Irradiation Configurations

A 3 cm thick brass collimator with divergent slits was used to generate arrays of planar minibeams. Different collimator geometries were investigated which were roughly based on the setup used in previous experiments at the Institute Curie in Orsay, France [12]. Table 1 summarizes the used slit dimensions, centre-to-centre spacing and divergence angles. The slit spacing increases from geometry *A* to *C* while the geometries *A*, *Ah* and *A3s* only differ in terms of slit height and number of slits, respectively.

**Table 1 cancers-15-00679-t001:** Dimensions of the considered collimator geometries. The slit widths ($w_{1-3}$), divergence angles (α,β) and centre-to-centre slit spacings (${\mathrm{ctc}}_{1,2}$) refer to the measures illustrated in Figure 1.

Label	No. of Slits	Slit Height [mm]	Slit Width [μm]			Divergence Angle [deg]		Spacing at Exit [μm]	
			$w_{1}$	$w_{2}$	$w_{3}$	α	β	${\mathrm{ctc}}_{1}$	${\mathrm{ctc}}_{2}$
**colli A**	5	10	400	400	425	0.595	0.632	1150	1275
**colli Ah**	5	5	400	400	425	0.595	0.632	1150	1275
**colli A3s**	3	10	400	400	-	0.595	-	1150	-
**colli B**	5	10	400	409	500	0.611	0.744	1900	1875
**colli C**	3	10	400	425	-	0.632	-	2425	-

Four different irradiation positions, labelled P1–P4, were considered (see Figure 2). P1 represents a central head-on irradiation where the minibeams traverse the lens and optical nerve before going through the skull into the brain. This corresponds to a worst-case scenario. P2 represents a peripheral head-on setup which would avoid direct irradiation of the macula and the optic nerve. Finally, P3 and P4 are two examples of angled irradiation configurations. The air gap between the collimator exit and the skin or cornea was 3 cm in all cases.

### 2.2. Dose Scoring and Analysis

Doses were scored in a sub-volume of the CT images using TOPAS’s *DoseToMedium* scorer. For the analysis, a 0.1 mm thick slice was selected from the centre of the dose distributions (see dashed yellow lines in Figure 2). The voxel size of these slices was 0.1 mm in the lateral direction and 0.5 mm along the longitudinal direction. From the dose distributions, the size and centre-to-centre distance of the minibeams were evaluated at different depths.

Different regions of interest (ROI) were delineated and the mean dose deposited in each ROI was calculated. Moreover, peak and valley doses, as well as the peak-to-valley dose ratio (PVDR), were evaluated at different depths. Due to the significant longitudinal variation in the dose distributions, it is challenging to state single peak and valley dose values that are representative for an entire ROI. Instead, peak and valley doses were assessed along lateral profiles at selected depths in the ROIs. The PVDR was computed accordingly. Figure 3 and Figure 4 indicate the positions of these lateral profiles and illustrate the considered ROIs.

The dose distributions were acquired by combining the results of 300 independent simulation instances. The dose uncertainty was assessed in each voxel by computing the standard deviation over these 300 instances. As a global uncertainty score, the relative mean uncertainty was computed, considering only voxels in the analysed region of the dose distribution. This was done to exclude very-low-dose regions which are irrelevant to the final dose analysis but which would have drastically increased and distorted the global uncertainty. The exact regions used for the global uncertainty score are indicated by the dashed yellow lines in Figure 5. The mean relative uncertainty was <4% in all cases (Table 2).

## 3. Results

The collimator geometries A, B and C produce slightly different minibeam patterns in terms of inter-beam spacing and minibeam size. The corresponding dose distributions are illustrated in Figure 6 and Table 3 lists the size of the central minibeam and the ctc distance between the central and adjacent minibeams at two different depths (tissue surface and target depth).

The main objective of this study was to evaluate the dose distributions of different example cases. For this, theoretical target regions were chosen that approximately correspond in size and location to typical intraocular tumour sites, such as uveal melanoma or retinoblastoma. Other considered ROIs were the part of the anterior chamber or skin (depending on the irradiation position) where the minibeam array enters the eye, the lens (if visible in the considered slice), the parts of the retina and skull that are traversed by the minibeam array and finally, an ROI covering the first few centimetres of the brain (except for P4), again limited to the region close to the minibeam array. As stated in the previous section, the exact location and size of all ROIs are illustrated in Figure 3 and Figure 4.

As a first step, the relative doses in the ROIs were considered, setting the mean dose in the target ROI at 100%. The mean dose across the ROIs and the peak and valley doses (as well as the corresponding PVDR) along selected lateral profiles were assessed and are compiled in Table 4. In the case of the brain/tissue ROIs, lateral profiles were considered at two positions to account for the greater longitudinal extension of the ROIs. The peak/valley doses and the PVDR are accordingly stated as a range.

In a second step, an example prescription dose of 30 Gy $D_{\mathrm{mean}}$ to the target was evaluated and the relative doses in Table 4 were converted to absolute doses. Table 5 compiles the corresponding results. The value of 30 Gy is a theoretical choice inspired by previous X-ray MBRT experiments with glioma-bearing rats where mean target doses of 28–30 Gy were prescribed [17,20]. For reference, proton therapy treatments for radioresistant uveal melanoma typically deliver 60–70 Gy in 4–5 fractions [1,21], while fractionation plans for stereotactic photon therapy may range from single fractions of 35 Gy (stereotactic radio *surgery*) [22] to 50–70 Gy delivered in as many as ten fractions (stereotactic radio *therapy*) [23].

## 4. Discussion

This study represents the first evaluation of orthovoltage X-rays MBRT as a potential alternative for the treatment of ocular tumours. The rationale for investigating this approach was twofold: (i) MBRT has already proven in several pre-clinical experiments its potential to increase normal tissue sparing while also providing high tumour control rates [8,9,17,20]. Consequently, the use of minibeams might help to further reduce radiation-induced adverse effects in OARs. (ii) Orthovoltage irradiators have a relatively small footprint and are comparatively cost-effective, which could favour widespread use. Implementations of X-ray MBRT at pre-clinical small animal irradiators have already been successfully demonstrated [9,12]. As there exist very similar clinical machines, a clinical translation could be expected to be relatively straightforward [12].

For this proof-of-concept study, we simulated the dose distributions for some example cases, including worst-case scenarios with a frontal irradiation field traversing the brain. Looking at the results in Table 4 and Table 5, one notices large dose depositions in several OARs (in particular, the skin/anterior chamber and lens, but also the bone) which may initially appear permissively high. However, due to the distinct spatial modulation of the dose in MBRT and the pronounced peak-to-valley dose differential, such high *peak* doses might still be tolerable and a direct comparison of the minibeam dose distributions with those of conventional RT may not be adequate or meaningful. In this context, it should also be noted that that the sparing effect of mini-/microbeam RT appears to depend primarily on the dose deposited in the *valley* regions [24]: For example, Dilmanian et al. [25] showed that the brain-sparing effect (measured by the onset of white matter necrosis) vanishes only when the valley dose approaches the tissue tolerance to broad beams.

Indeed, numerous previous studies highlighted the different biological responses observed after MBRT or conventional broad beam irradiation, both in terms of normal tissue tolerance and tumour control [9,12,20,26]. Some relevant results from selected in vivo experiments are compiled in Table 6. This experimental MBRT data can be used in combination with the available broad beam toxicity data to try to put the simulation results into perspective.

Shallow structures such as the lens, eyelashes and the lacrimal system are regarded to be radiosensitive [21,27]. Finger [27] reports that eyelids react like (thin) normal skin and that moist desquamation may occur at 50–60 Gy delivered in 1.8–2 Gy fractions. Loss of eyelashes may start at 10 Gy and can become permanent at 30 Gy. In the 30 Gy examples presented in Table 5, the skin/anterior chamber doses are 40–50 Gy mean and 90–160 Gy in the peaks. On the other hand, Bazyar et al. found that normal mouse skin could well tolerate peak doses of up to 150 Gy (using however roughly threefold smaller minibeams) [9]. It may be noted that the skin of the eyelids and the cornea respond differently to irradiation and Finger states that “most acute corneal toxicity results from a loss of the tear film with secondary keratitis sicca” [27] and thus is more influenced by damage to the lacrimal system or conjunctiva, which were outside of the irradiation field in the simulated examples. Moreover, as stated above, the most relevant quantity for tissue sparing appears to be the valley dose which was well below 10 Gy in all but one of the considered examples.

For the lens, Finger reports that “as little as 2 Gy in a single fraction or 8 Gy in multiple fractions can induce cataract” [27] and Desjardins et al. state that “the lens is the most radiosensitive tissue” in the eye [21]. In this regard, it may be preferable, if possible, to choose a configuration such as P2 where direct irradiation of the lens can be avoided. The currently available minibeam data, however, do not allow for a direct comparison.

**Table 6 cancers-15-00679-t006:** Compilation of selected in vivo studies evaluating orthovoltage X-ray MBRT (n.r. = not reported).

Study	Model	Configuration	FWHM/ctc [mm]	Mean/Peak/Valley Dose [Gy]	Results/Observations
Bazyar et al., 2017 [9]	normal mouse skin	single array	0.25/0.93	n.r./150/∼6.5	no radiation side effects
	mouse melanoma model	single array	0.25/0.93	n.r./150/∼6.5	MBRT more effective than conv. RT (slower growth rate, longer mean survival)
Bertho et al., 2022 [20]	glioma-bearing rat brain	single array	0.7/1.4	30/83/4.5	33% long-term survival; no skin toxicity (immunocompetent rats)
Dilmanian et al., 2006 [28]	normal rat spinal cord	single array	0.68/4	n.r./400/n.r.	irradiation tolerated long-term by 3/4 rats; lag in weight gain with respect to unirradiated controls
Deman et al., 2012 [8]	normal rat brain	single array	0.62/1.22	n.r./123/∼4.1	no clinical alteration or MRI images abnormalities
	glioma-bearing rat brain	two arrays, interleaved	0.62/1.22	54 (homog. in target)	significantly increased survival with respect to untreated controls
Prezado et al., 2012 [29]	glioma-bearing rat brain	single array	0.64/1.12	n.r./180/16	no benefit with respect to untreated controls
	glioma-bearing rat brain	two arrays, interleaved	0.64/1.12	70–100 (homog. in target)	significantly increased survival with respect to untreated controls
Prezado et al., 2015 [7]	normal rat brain	single array	0.6/1.2	n.r./100/6.6	alive 560 days after irradiation, normal behaviour; signs of haemorrhage, small vascular damage; microcalcifications in histological analysis
Prezado et al., 2017 [12]	normal rat brain	single array	0.97/1.61	20/58/4.7	no brain damage in whole-brain irradiation
Sotiropoulos et al., 2021 [17]	glioma-bearing rat brain	single array	0.70/1.47	28/81/7.2	significantly increased survival with respect to untreated controls

The mean dose to the retina ROI was around 25±2 Gy in all cases in the presented examples with valley dose ranging from 3 to 11 Gy. While Finger states that “doses as little as 18 Gy can induce radiation retinopathy in patients with compromised chorioretinal circulation”, he generally sees a moderate risk only for doses >35 Gy [27]. Moreover, Desjardins et al. state that “the retina and optic disk are not very radiosensitive but most of the radiation side effects are due to endothelial vascular damage causing vascular occlusions” [21] and a study by Parsons et al. including 131 patients found no damage in the optic nerve for doses <59 Gy and a 15-year actuarial risk of 11% for optic neuropathy only at doses ≥60 Gy [30].

Concerning the ROI in the bone, mean doses were found to be about 5–40% higher than the mean target dose in all cases except *P2, colli Ah* (the traversed part of the skull appears to be less thick). Moreover, rather high valley doses in the centre of the bone in the order of 10–20 Gy are observed in basically all considered examples. The corresponding example doses in the brain ROI were $D_{\mathrm{mean}}\leq 40\%$ with valley doses mostly below 7 Gy. To put this into perspective, one may look at the values reported by Bertho et al. [20], Deman et al. [8], Prezado et al. [12] and Sotiropoulos et al. [17] (see Table 6). All of these studies reported mean and peak doses in the brain that are significantly higher than those in the examples considered here and all studies found these doses to be well tolerated. As the irradiation was performed through the skin and skull, one might argue that the high peak doses in the skull were also well tolerated (although a caveat may be that the PVDR in the skull was likely higher in the animal studies as a result of the shallower depths). Moreover, in an article on fractionated radiosurgery for orbital and ocular tumours [31], Morales et al. present examples of treatment plans with dose depositions surpassing 40% in sizeable portions of the brain and skull. These dose distributions were furthermore laterally homogeneous, whereas a notable modulation of the dose (PVDR >2) was maintained in the minibeam examples, which could improve tissue tolerance.

Finally, it might also be interesting to highlight the positive results of the re-challenging study performed by Bertho et al. [20]: Glioma-bearing rats that had been cured through minibeam or conventional irradiation were re-challenged with a second tumour implantation. It was shown that none of the previously irradiated animals developed a macroscopic tumour, suggesting a possible anti-tumour immunity following a successful high-dose treatment as feasible with minibeams. These results could be promising in the context of metastases which occur for 20–30% of uveal melanoma patients treated with proton therapy [21].

The MC study presented here has a few limitations which relate to (i) the setup of the simulations and (ii) the conclusions that can be drawn from the results. Concerning the first point, it should be noted that the simulations were performed for theoretical example cases, using generic target volumes. While we think that such an approach is justified and adequate for a proof-of-concept study, as presented here, future studies could evaluate concrete patient cases and include comparisons with the dose distributions of established treatment plans. Moreover, next to the single minibeam arrays simulated here (which are very common in pre-clinical MBRT studies), multi-array configurations might be investigated.

Secondly, due to the lack of biological data describing the effect of orthovoltage X-ray MBRT on ocular or orbital OARs, it is difficult, at this stage, to draw specific conclusions from the simulated dose distributions. Ocular radiation toxicity data are only available for conventional broad beam irradiations which, as argued above, may only have limited applicability in the context of minibeams. Nonetheless, comparison of the simulated example doses (Table 5) with the existing data on X-ray MBRT (Table 6) does generally paint a promising picture. We therefore think that these results encourage the realisation of dedicated animal studies considering minibeam irradiations of the eye.

## 5. Conclusions

MBRT represents a very promising new therapeutical approach that has already proven, in several pre-clinical experiments, its potential to simultaneously increase normal tissue sparing and provide tumour control comparable or even superior to that of conventional RT [8,9,20,29]. In this study, we performed a first in silico evaluation of the use of this technique for the treatment of ocular tumours. To this end, dose distributions in CT images of a human head were simulated and doses in various ROIs were assessed. Due to the requirement of beam energies in the lower orthovoltage range, X-ray MBRT is particularly well suited for implementation at small and cost-effective irradiators [12]. Provided that an iso-efficacy can be achieved, this aspect could represent an important advantage over established high-precision techniques such as proton therapy, Gammaknife or CyberKnife (i.e., stereotactic RT), which are suffering from both high costs and relatively low availability.

An important limitation for the interpretation of the simulation results is the absence of experimental biological data considering the irradiation of ocular structures with X-ray minibeams. Moreover, reference values and models developed for conventional RT techniques (such as the *linear-quadratic model*) are probably not adequate for application to MBRT cases either [32]. It is therefore, at the moment, not possible to draw specific conclusions from the simulated dose distributions.

On the other hand, the comparison of the example doses in Table 5 with the doses reported in previous X-ray MBRT animal experiments (Table 6) appears encouraging and supports, in our opinion, further investigations. In this context, more insight might be gained in particular from dedicated in vivo studies, comparing the eye toxicities of conventional broad beam irradiation and MBRT and evaluating tumour control, for instance, in a rat melanoma model.

## Figures and Tables

**Figure 1 cancers-15-00679-f001:**
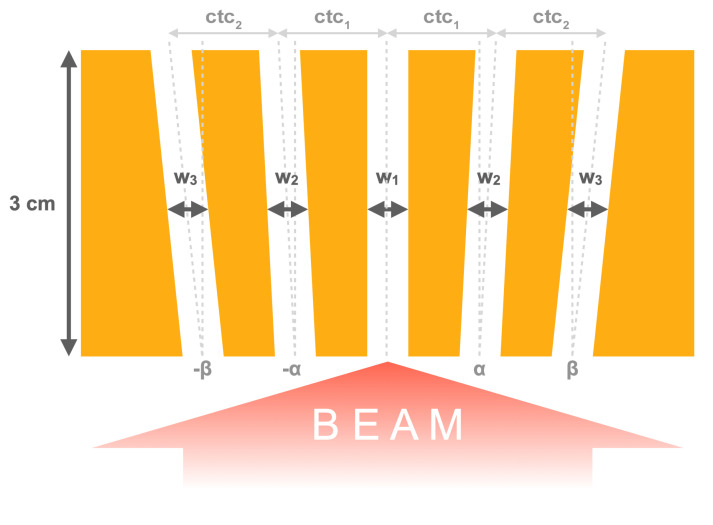
Schematic of the collimator used for minibeam generation illustrating the slit widths and divergence angles listed in Table 1.

**Figure 2 cancers-15-00679-f002:**
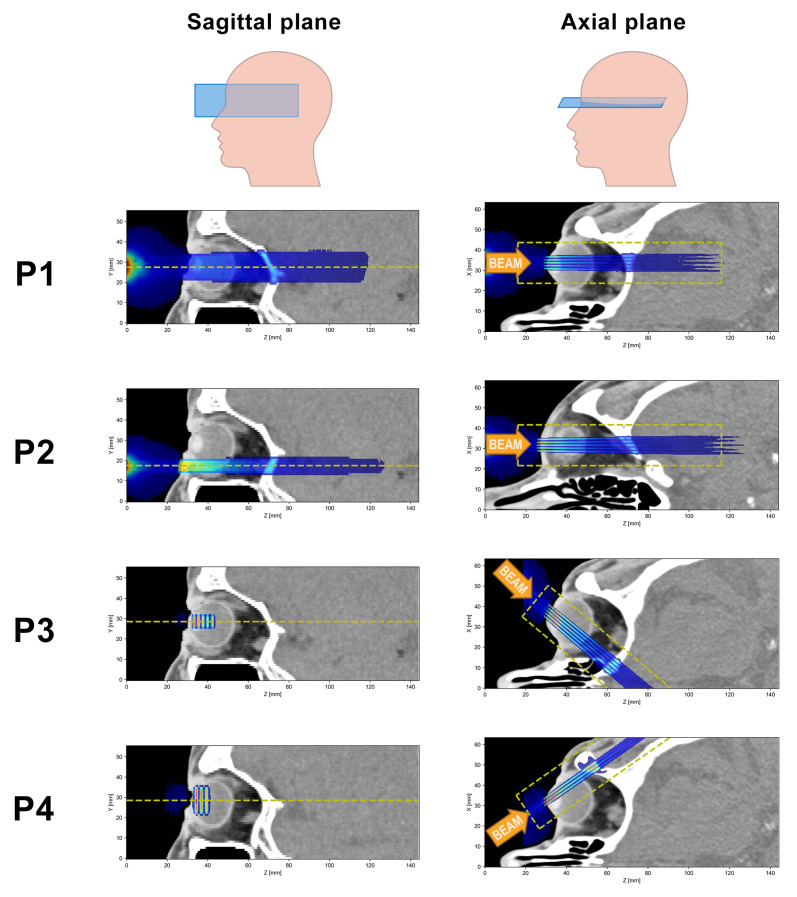
Illustration of the four different irradiation positions (P1–P4): For each row, the left and right panels show sagittal and axial sections, respectively, of the same dose distribution, while the orange arrows indicate the beam direction. The dashed yellow lines in the left panels indicate the position of the dose slices mentioned in Section 2.2, while the dashed yellow rectangles in the right panels outline the region within the slices that were considered in the dose analysis.

**Figure 3 cancers-15-00679-f003:**
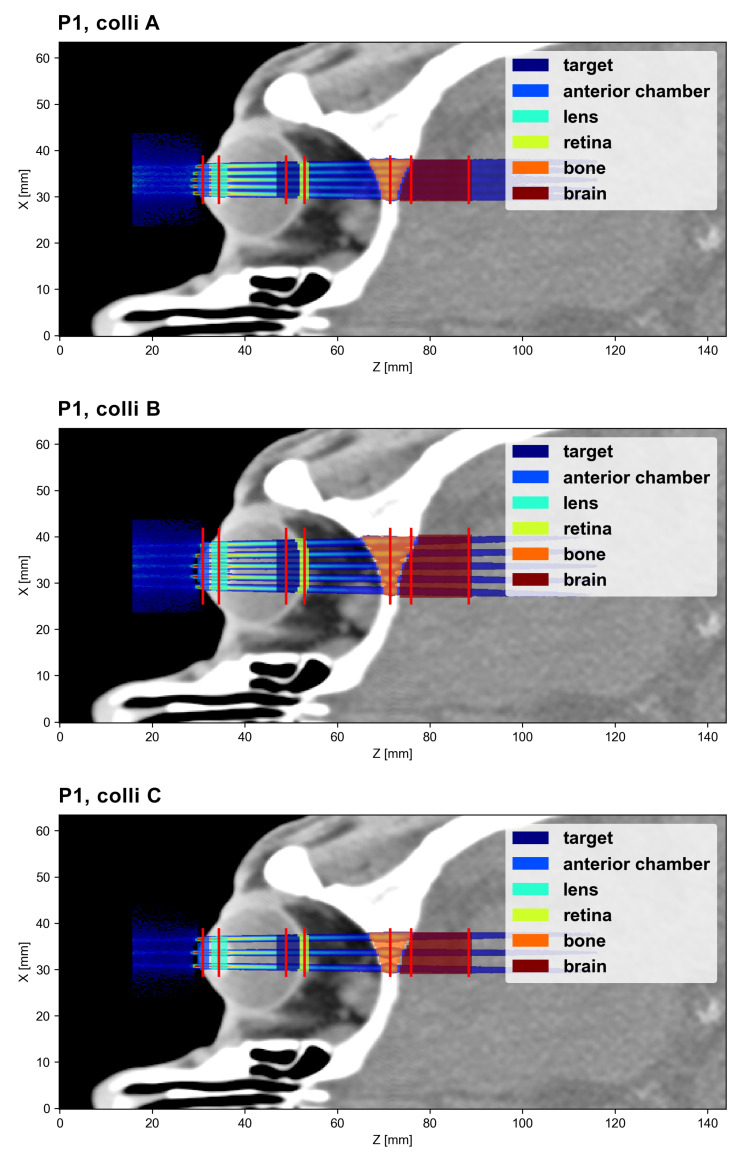
Dose analysis for the three P1 cases: The ROIs are shown and the sampling positions of the lateral profiles are indicated (red lines).

**Figure 4 cancers-15-00679-f004:**
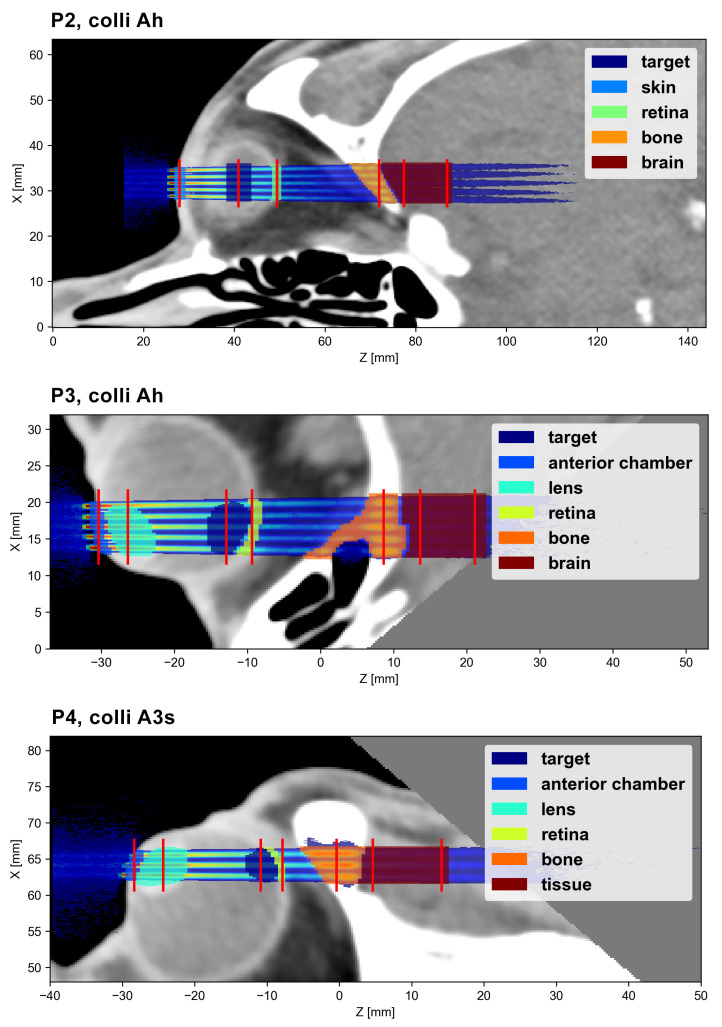
The ROIs and the lateral profile sampling positions (red lines) for the P2, P3 and P4 cases.

**Figure 5 cancers-15-00679-f005:**
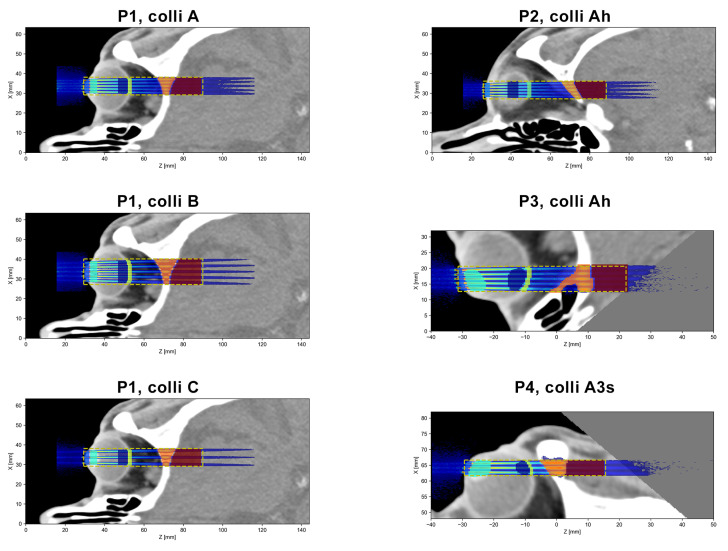
Regions of the dose distributions used for global uncertainty calculation. Only the voxels inside the regions delineated by the yellow dashed lines were considered.

**Figure 6 cancers-15-00679-f006:**
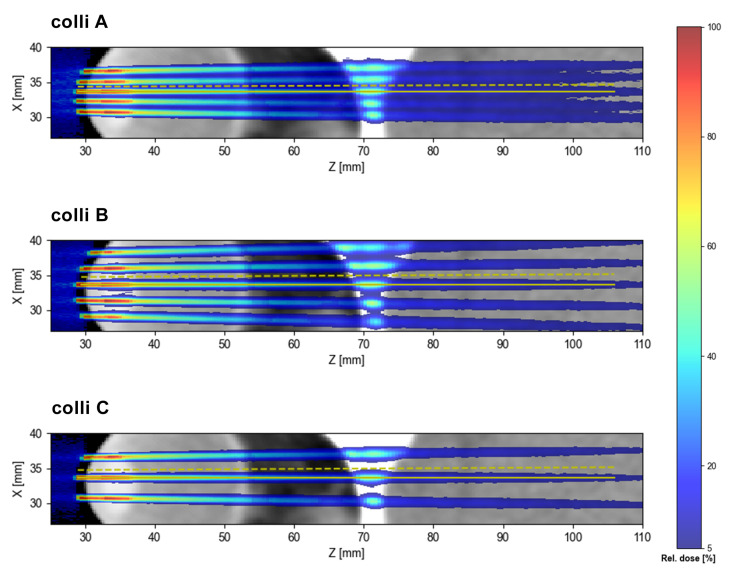
Minibeam dose patterns obtained with the three collimator geometries A, B and C.

**Table 2 cancers-15-00679-t002:** Global relative mean dose uncertainties for the different cases.

Case	Global Relative Mean Dose Uncertainty
P1, colli A	2.4%
P1, colli B	2.7%
P1, colli C	3.8%
P2, colli Ah	2.6%
P3, colli Ah	2.3%
P4, colli A3s	1.9%

**Table 3 cancers-15-00679-t003:** Full width at half maximum (FWHM) and spacing between central minibeams stated as centre-to-centre (ctc) distance for the different collimator geometries in the P1 position. The depth refers to the depth in tissue.

Position	Depth [cm]	FWHM/ctc [mm]		
		Colli A	Colli B	Colli C
Surface	0	0.69/1.3	0.64/2.2	0.64/2.9
Target	2	0.90/1.5	0.86/2.4	0.85/3.1

**Table 4 cancers-15-00679-t004:** Mean, peak and valley doses in the ROI for the different irradiation cases relative to the mean target dose.

Case	Volume	$D_{\mathrm{mean}}$ [%]	$D_{\mathrm{peak}}$ [%]	$D_{\mathrm{valley}}$ [%]	PVDR
**P1, colli A**	target	100	164	37	4.5
	anterior chamber	163	327	26	12.8
	lens	179	338	35	9.8
	retina	83	138	36	3.8
	bone	107	193	71	2.7
	brain	37	61–44	26–22	2.3–2.0
**P1, colli B**	target	100	227	18	12.6
	anterior chamber	157	468	16	30.2
	lens	176	475	24	20.0
	retina	86	194	15	13.4
	bone	108	265	33	7.9
	brain	38	83–62	11–10	7.8–6.4
**P1, colli C**	target	100	251	13	19.3
	anterior chamber	154	527	12	44.6
	lens	180	533	18	29.0
	retina	85	215	11	19.8
	bone	108	288	25	11.6
	brain	38	90–67	6–6	14.7–10.4
**P2, colli Ah**	target	100	196	29	6.9
	skin	162	341	21	16.6
	retina	77	129	27	4.9
	bone	86	157	47	3.4
	brain	33	50–39	22–19	2.3–2.0
**P3, colli Ah**	target	100	174	29	6.0
	anterior chamber	162	363	19	19.6
	lens	165	357	27	13.2
	retina	87	154	28	5.4
	bone	101	222	64	3.5
	brain	40	60–47	22–20	2.7–2.4
**P4, colli A3s**	target	100	175	29	6.0
	anterior chamber	155	336	17	19.9
	lens	141	358	24	15.1
	retina	88	157	30	5.2
	bone	141	276	67	4.1
	tissue	48	90–66	28–25	3.2–2.6

**Table 5 cancers-15-00679-t005:** Mean, peak and valley doses in the ROI for the different irradiation cases, assuming a prescription target dose of $D_{\mathrm{mean}}=30$ Gy.

Case	Volume	$D_{\mathrm{mean}}$ [Gy]	$D_{\mathrm{peak}}$ [Gy]	$D_{\mathrm{valley}}$ [Gy]	PVDR
**P1, colli A**	target	30.0	49.1	11.0	4.5
	anterior chamber	48.8	98.2	7.7	12.8
	lens	53.5	101.4	10.4	9.8
	retina	24.8	41.4	10.8	3.8
	bone	32.0	57.8	21.3	2.7
	brain	11.2	18.2–13.2	7.8–6.5	2.3–2.0
**P1, colli B**	target	30.0	68.2	5.4	12.6
	anterior chamber	47.1	140.4	4.7	30.2
	lens	52.6	142.5	7.1	20.0
	retina	25.7	58.1	4.3	13.4
	bone	32.4	79.5	10.0	7.9
	brain	11.3	24.9–18.5	3.2–2.9	7.8–6.4
**P1, colli C**	target	30.0	75.3	3.9	19.3
	anterior chamber	46.3	158.0	3.5	44.6
	lens	54.0	159.8	5.5	29.0
	retina	25.4	64.6	3.3	19.8
	bone	32.5	86.4	7.5	11.6
	brain	11.3	27.0–20.0	1.8–1.9	14.7–10.4
**P2, colli Ah**	target	30.0	58.9	8.5	6.9
	skin	48.7	102.3	6.2	16.6
	retina	23.2	38.8	7.9	4.9
	bone	25.7	47.2	14.0	3.4
	brain	9.8	15.1–11.7	6.5–5.8	2.3-2.0
**P3, colli Ah**	target	30.0	52.1	8.7	6.0
	anterior chamber	48.5	108.9	5.6	19.6
	lens	49.6	107.0	8.1	13.2
	retina	25.9	46.2	8.5	5.4
	bone	30.3	66.6	19.3	3.5
	brain	11.9	18.1–14.2	6.6-6.0	2.7–2.4
**P4, colli A3s**	target	30.0	52.4	8.7	6.0
	anterior chamber	46.6	100.8	5.1	19.9
	lens	42.4	107.5	7.1	15.1
	retina	26.3	47.0	9.0	5.2
	bone	42.2	82.8	20.0	4.1
	tissue	14.3	26.9–19.8	8.3–7.6	3.2–2.6

## Data Availability

The data presented in this study are available upon reasonable request to the corresponding author.

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
