# Peer review of "Orthovoltage X-ray Minibeam Radiation Therapy for the Treatment of Ocular Tumours—An In Silico Evaluation"

_cancers, 2023, doi:10.3390/cancers15030679_

Round 1

Reviewer 1 Report

The work under review constitutes a proof of concept, using Monte Carlo simulation, for the use of orthovoltage X-ray minibeam radiotherapy (MBRT) to treat ocular tumors. The topic of MBRT is progressively gaining scientific interest, due mainly to the highly promising preliminary results obtained from small animal treatments, both in terms of disease local control and normal tissue sparing, thus warranting further investigation.

The manuscript is overall well written, comprehensive, and easy to follow. The scientific methods are sound, and results are adequately presented and discussed. The authors are commended for their work. Following are some minor comments/suggestions they may find useful to construct a final version of their manuscript.

1) Abstract, line 13: Consider replacing “excellent results” with “increased local control rates”, since “results” include both local control and normal tissue toxicity.

2) Introduction, line 74: There are two “while” there.

3) Figure 2 legend: In the case of axial plots, it seems that the yellow lines indicate ROIs and not slices. 

4) Figure 6: It would be nice to include a colorbar to allow for quantitative inspection of the displayed dose distributions.

Author Response

REVIEWER 1

We are very grateful for the reviewer’s comments and suggestions. As a result, we have made a few revisions to the text which are indicted in red in the manuscript.

1) Abstract, line 13: Consider replacing “excellent results” with “increased local control rates”, since “results” include both local control and normal tissue toxicity.

Answer to reviewer: Replaced as suggested.

2) Introduction, line 74: There are two “while” there.

Answer to reviewer: Corrected.

3) Figure 2 legend: In the case of axial plots, it seems that the yellow lines indicate ROIs and not slices.

Answer to reviewer: We thank the reviewer for this observation. In fact, the single yellow line in the left panels indicates the position of the considered dose slice while the yellow rectangles in the right panels indicate indeed the considered ROIs (however, the term "ROI" was used in this manuscript to refer to contoured ares/organs of risks).

The figure legend has been updated to be more precise: "The dashed yellow lines in the left panels indicate the position of the dose slices mentioned in section 2.2 while the dashed yellow rectangles in the right panels outline the region within the slices that were considered in the dose analysis."

4) Figure 6: It would be nice to include a colorbar to allow for quantitative inspection of the displayed dose distributions.

Answer to reviewer: We thank the reviewer for the suggestion. A colorbar indicating the relative dose levels has been added.

Reviewer 2 Report

The study is well written and represents the evaluation of orthovoltage X-rays MBRT as a potential alternative for the treatment of ocular tumors. It is of high interest to the radiotherapy community, though it is still a proof-of-concept.

Nevertheless, there are few remarks/suggestions:

1. This is a MC study. How did you validate the MC model?

2. The last sentence of first para in the INTRO section ( In this context, minibeam...) comes suddenly. At that moment the reader is not aware of MBRT so it is not clear why MBRT is promising approach. Please remove the sentence bellow or rewrote it accordingly.  

3. The last sentence in the INTRO section. Yes, this is a first MC simulation of MBRT ocular tumors. But, there a some other MC simulations of MBRT. It would be clearer if the 'in this context' in the sentence be replaced with 'of ocular tumors'?

4. Figure 2 should be presented without dose distribution on the images. Only the beam directions. Figure should be better explained. One could not easily understand what is the first P1 and what is the second P1 image.

Author Response

REVIEWER 2

We are very grateful for the reviewer’s comments and suggestions. As a result, we have made a few revisions to the text which are indicted in red in the manuscript.

1. This is a MC study. How did you validate the MC model?

Answer to reviewer: The MC model of the MBRT implementation at the SARRP had been developed and already validated in previous studies of our team. A very good agreement between simulated and measured dose distributions is reported in Prezado et al., 2017 (doi: 10.1038/s41598-017-17543-3) and further validation results are also presented in Gonzalez et al., 2020 (doi: 10.1016/j.ejmp.2019.12.016). The MC simulations in those studies were performed using "pure" Geant4 whereas this study used TOPAS. However, the physics simulation core of TOPAS is Geant4 which is why the aforementioned validation results can be considered applicable also to this study.

An according sentence has been added to the first paragraph of section 2.

2. The last sentence of first para in the INTRO section ( In this context, minibeam...) comes suddenly. At that moment the reader is not aware of MBRT so it is not clear why MBRT is promising approach. Please remove the sentence bellow or rewrote it accordingly.  

Answer to reviewer: We thank the reviewer for this suggestion. The transitions between the first two paragraphs in the introduction section has been improved. New text: "…which may entail a considerable impairment of visual acuity. ###END PARAGRAPH 1### Minibeam radiation therapy (MBRT) is a promising new treatment approach that in preclinical studies has shown to improve normal tissue sparing while maintaining high tumor control rates [6]. In the context of ocular tumours, it could help to further reduce side effects in the aforementioned organs at risk (OARs) and to better preserve the patient’s vision."

3. The last sentence in the INTRO section. Yes, this is a first MC simulation of MBRT ocular tumors. But, there a some other MC simulations of MBRT. It would be clearer if the 'in this context' in the sentence be replaced with 'of ocular tumors'?

Answer to reviewer: Corrected as suggested.

4. Figure 2 should be presented without dose distribution on the images. Only the beam directions. Figure should be better explained. One could not easily understand what is the first P1 and what is the second P1 image.

Answer to reviewer: One main purpose of Figure 2 is to illustrate which regions of the dose depositions were considered in the analysis. For this, we think that it is indispensable to actually display the dose distributions.

In order to improve the clarity of the figure, we have added arrows indicating the beam directions and we updated the figure caption.